# CuI-Catalyzed Coupling Reactions of 4-Iodopyrazoles and Alcohols: Application toward Withasomnine and Homologs

**DOI:** 10.3390/molecules26113370

**Published:** 2021-06-02

**Authors:** Yoshihide Usami, Yumika Kubo, Toshiki Takagaki, Nao Kuroiwa, Jun Ono, Kohei Nishikawa, Ayaka Nakamizu, Yuya Tatsui, Shinya Harusawa, Noboru Hayama, Hiroki Yoneyama

**Affiliations:** Department of Pharmaceutical Organic Chemistry, Osaka University of Pharmaceutical Sciences, 4-20-1 Nasahara, Takatsuki, Osaka 569-1094, Japan; e12343@gap.oups.ac.jp (Y.K.); e16442@gap.oups.ac.jp (T.T.); e16332@gap.oups.ac.jp (N.K.); e15439@gap.oups.ac.jp (J.O.); e15637@gap.oups.ac.jp (K.N.); e14519@gap.oups.ac.jp (A.N.); e18902@gap.oups.ac.jp (Y.T.); harusawa@gly.oups.ac.jp (S.H.); hayama@gly.oups.ac.jp (N.H.); yoneyama@gly.oups.ac.jp (H.Y.)

**Keywords:** synthesis, 4-alkoxypyrazole, CuI, coupling reaction, microwave, withasomnine, homologue

## Abstract

The direct 4-alkoxylation of 4-iodo-1*H*-pyrazoles with alcohols was achieved by a CuI-catalyzed coupling protocol. The optimal reaction conditions employed excess alcohol and potassium *t*-butoxide (2 equiv) in the presence of CuI (20 mol%) and 3,4,7,8-tetramethyl-1,10-phenanthroline (20 mol%) at 130 °C for 1 h under microwave irradiation. The present method was efficiently applied to the synthesis of withasomnine and its six- and seven-membered cyclic homologs.

## 1. Introduction

Owing to their diverse bioactivities, both natural and synthetic pyrazoles and pyrazole-fused heterocycles have been widely exploited as pharmaceutical or pesticide active ingredients [1,2,3,4]. Therefore, the efficient synthesis of substituted pyrazoles possessing characteristic functionalities at specific positions is an important objective in organic and medicinal chemistry, as well as in drug discovery. In this context, we recently reported palladium- or copper-catalyzed C–N coupling reactions at the C-4 positions of pyrazoles [5]. Although metal-catalyzed C–O coupling reactions have been widely reported, owing to their wide-ranging potentials [6,7,8,9,10,11,12], the direct C4-*O*-functionalization of pyrazoles has not yet been studied satisfactorily [13,14] despite the important bioactivities that have been demonstrated for several 4-alkoxypyrazoles, as presented in Figure 1.

4-Hyroxypyrazole, a metabolite of pyrazole, exhibits various biological activities such as anti-inflammatory, antipyretic, antitumor, antifungal effects [15]. 4-Methoxy-, 4-ethoxy-, 4-*n*-propoxy-, and 4-isopropoxypyrazoles have been reported to inhibit liver alcohol dehydrogenase (LAD) in humans, rats, and horses [16]. In particular, 4-ethoxypyrazole and 4-propyloxypyrazole have been recognized as a cytochrome P-450 inducer [17] and cytochrome P450 2E1 inhibitor, respectively [18]. Two C4-*O*-functionalized pyrazoles have been patented: 1-methyl-3,5-diphenyl-4-propoxypyrazole, as a fungicide [19], and the alkyl (4-alkoxy-1-phenyl)pyrazolylcarboxylates, which possess antipyretic, sedative, anti-inflammatory, and analgesic activities [20]. A compound bearing a 4-(2,4-difluorophenyl)oxy group was revealed as a human dihydroorotate dehydrogenase (DHODH) inhibitor [21].

4-Allyloxy-1*H*-1-tritylpyrazole **4a** (Scheme 1), derived from 4-iodopyrazole (**1**), played a key role as a versatile intermediate in our previous studies for the total synthesis of the pyrazole alkaloid, withasomnine (**7**) [22,23], and its six-membered homolog **11** [23,24,25,26,27], which were reported to exhibit COX-2 inhibitory activities [23,27,28]. Compound **4a** was also extensively utilized as an important intermediate for the construction of new pyrazole-fused heterobicyclic molecules **9** via ring-closing metathesis (RCM) (Scheme 1) [29,30,31]. However, the synthesis of compound **4a** requires six steps from commercially available pyrazole, through 4-iodopyrazole (**1**), 4-iodo-1*H*-1-tritylpyrazole (**2a**), and aldehyde **3**. If direct *O*-allylation from **1** could be realized, the synthesis of several synthetic targets would be remarkably shortened. Thus, we focused our attention on the direct C–O coupling reactions of **1**, based on our prior report of the C–N couplings of 4-halopyrazoles [5]. Herein, we disclose CuI-catalyzed coupling reactions of 4-iodo-1*H*-pyrazoles and alcohols. Furthermore, the developed method was applied to improve the synthesis of withasomnine and its homologs containing six- or seven-membered ring systems.

## 2. Results and Discussions

### 2.1. Investigation of 4-O-Allylation of 4-Iodopyrazole 

Initially, we attempted the Pd(dba)_2_-catalyzed reaction between **2a** and allyl alcohol (2 equivalent (equiv)) in the presence of *t*BuDavePhos as a ligand and potassium *tert*-butoxide (*^t^*BuOK) as a base under the reaction conditions in our previous report [5]; however, none of the desired coupling product was obtained (Table 1, entry 1). The corresponding 4-bromo-1*H*-1-tritylpyrazole was not effective in the palladium-catalyzed coupling reaction. Then, the CuI-catalyzed reaction between 4-iodo-1*H*-1-tritylpyrazole **2a** and allyl alcohol was examined; the results are summarized in Table 1. All reactions were performed using **2a** (50 mg) in a solvent (2.0 mL). In the presence of ligand 2-isobutyroylcyclo-hexanone (**L2**) or 1,10-phenanthroline (**L3**) in *N,N*-dimethylformamide (DMF) [5], reactions of **2a** and allyl alcohol (2 equiv) did not afford **4a** (entries 2 and 3, respectively). However, when allyl alcohol was used as a solvent for this reaction with **L3** at 100 °C overnight, the desired C4-*O*-allylation product **4a** was obtained in 51% yield (entry 4). Next, microwave (MW) assistance was applied to reduce the reaction time (entries 5–9). In these experiments, the reaction time was fixed at 1 h and the ligand was changed to 3,4,7,8-tetramethyl-1,10-phenanthroline (**L4**). From entry 6, the optimum reaction temperature was determined to be 130 °C, giving **4a** in 66% yield. At 160 °C, the reaction mixture turned black with a poor yield of **4a** (16%, entry 8). In addition, shortening the reaction time (30 min) or reducing the amount of CuI to 10 mol% afforded **4a** in lower yields (entry 7:24%; entry 9:37%, respectively). Based on these results, the optimum conditions obtained in entry 6 were applied in the following coupling reactions of 4-iodopyrazoles with various alcohols.

### 2.2. C4-Alkoxylation of 4-iodopyrazole with Alcohols Using CuI-Catalyzed Coupling 

To study the scope and limitations of this transformation, the CuI-catalyzed reactions of iodopyrazoles **2** (50 mg) with various alcohols (2.0 mL, excess amount) were carried out under the optimal conditions (Table 1, entry 6). The results are summarized in Table 2. The reactions of **2a** with linear short-chain primary alcohols (methanol, ethanol, and *n*-propanol) afforded the corresponding products **4c**, **4d**, and **4e** in moderate yields (61–76%, entries 1–3), while the reaction with a longer-chain primary alcohol (*n*-butanol) resulted in a lower yield (33%, entry 4). The reactions of **2a** with branched primary alcohols (isobutyl and isoamyl alcohols) provided **4i** (45%) and **4k** (37%), respectively (entries 7 and 9), but with secondary isopropanol, gave **4g** in only 9% yield (entry 5). The presence of *sec*- or *tert*-butyl groups in the alcohol was not compatible with the present reaction conditions (entries 6 and 8), probably due to steric hindrance. In contrast, the reactions with cyclic secondary alcohols did proceed (entries 10, 11, and 12), but the respective isolated yields of the coupled products **4l**, **4m**, and **4n** were 59%, 18%, and 25%, respectively. In these reactions, 1.0 mL of cyclic alcohol was used with respect to substrate **2a** (50 mg); the high boiling points (cyclobutanol: 123 °C/733 mmHg; cyclopentanol: 139–140 °C; cyclohexanol: 160–161 °C) of these materials complicated product isolation by chromatography. Furthermore, when 2 equivalents of the cyclic alcohols and acetonitrile (2.0 mL) as a co-solvent were used, no coupled products could be detected. Although the reaction with benzyl alcohol (bp: 205 °C) was also difficult, the use of benzyl alcohol (1.0 mL) and toluene as a co-solvent (1.0 mL) afforded the corresponding product (**4o**) in poor yield (12%, entry 13). With phenols, no desired coupling products were obtained under various reaction conditions (entries 14 and 15). In the case of p-methoxyphenol (entry 15), a detailed analysis of the reaction mixture revealed a trace amount of 5,5′-dimethoxy-2,2′-biphenyldiol, which has been reported to have radical scavenging or antibacterial activities [32,33]. The initially formed dihydroxybiphenyls [34] might inhibit the attempted C–O coupling reaction. 

The direct introduction of the allyloxy group at the C4 position of *N*-alkenyl-4-iodo-1*H*-pyrazoles (**2b**, **2c**, **2d**) by CuI-mediated reaction afforded the expected products **4r**, **4s**, and **4t** in moderate yields (entries 17–19). These products were subsequently applied in the synthesis of withasomnine and its analogs (Scheme 2). Neither the C–O coupling reaction of **2a** with water nor of *N*-nonprotected iodopyrazole **1** with allyl alcohol was successful (entries 20 and 21). 

In preliminary experiments, the Pd(dba)_2_-catalyzed coupling of **2a** with four types of alcohols (methanol, ethanol, *n*-propanol, and *tert*-butyl alcohol) under the same conditions as mentioned above was examined; however, these trials did not give the desired coupling products, yielding only hydrogenated 1*H*-1-tritylpyrazole in 52, 63, 64, and 8% yields, respectively. 

### 2.3. Application to Improved Synthesis of Withasomnine and Six- and Seven-Membered Cyclic Homologs

A modified synthesis of withasomnine and its homologs using the products described above was performed to demonstrate the usefulness of the present method. The improved synthesis of withasomnine (**7**) is summarized in Scheme 2. 4-Iodo-1*H*-pyrazole (**1**) was treated with allyl bromide under basic conditions to give *N*-allylated compound **2c** in 97% yield. The double bond in the *N*-allyl group in **2c** was migrated by treatment with a ruthenium hydride catalyst (RuClH(CO)(PPh_3_)_3_) to give an *E/Z* mixture of **2b** in 96% yield, which was transformed to **4r** by CuI-catalyzed coupling, as described above (Table 2, entry 17). The Claisen rearrangement of **4r** gave *(E/Z)-***12** (87%), which was subsequently *O*-triflated by treatment with trifluoromethanesulfonic anhydride (Tf_2_O) in the presence of triethylamine at −20 °C to yield ring-closing metathesis (RCM) substrate **13** in 90% yield. Treatment of **13** with Grubbs^2nd^ catalyst in toluene at 100 °C under MW irradiation gave the desired RCM product **14** in 0–58% yields with unsatisfied reproducibility. 

Alternatively, CuI-assisted RCM [24,35] of **13** in CH_2_Cl_2_ under milder conditions using MW-aided heating at 80 °C for 1 h successfully afforded pyrrole-[1,2-*b*] pyrazole **14** (63%), which was immediately hydrogenated under a hydrogen gas atmosphere with Pd/C in MeOH to give penultimate product **6** in 90% yield. As the transformation from **6** to **7** via a Suzuki-Miyaura coupling reaction has already been reported [22,23], the present approach constitutes a formal total synthesis of withasomnine (**7**). The overall yield of **7** in this case was 24% over nine steps from commercially available pyrazole, whereas that of our previous method was 8% in 13 steps. Therefore, the current synthesis realizes a four-step reduction and nearly threefold improvement in overall yield [22,23].

The syntheses of the six-and seven-membered cyclic homologs **11** and **15** are summarized in Scheme 3. The total yield of **11** was improved by ~1.6-fold over our former synthesis based on the yields of transformations from **1** to **2c** (seen in Scheme 2) and **2c** to **4s** (Table 2, entry 18) [24]. Our synthesis of another withasomnine homolog, **15**, previously achieved by Allin via radical cyclization [25,26], began with the transformation of **1** to **2d** in 88% yield. Compound **2d** was *O*-allylated using the present method to **4t**, as described previously (Table 2, entry 19). Then, **4t** was rearranged into **16** (81% yield) under MW-assisted heating, and subsequent *O*-triflation afforded **17** (83% yield). RCM substrate **17** was similarly cyclized to seven-membered intermediate **18** in 72% yield, which was then subjected to Suzuki-Miyaura coupling with phenylboronic acid to afford **19** in 87% yield. The synthesis of **15** was completed in 92% yield by the Pd-C-catalyzed hydrogenation of **19**. An alternative route to **15** comprised the transformation of **1** to **4b** in 52% yield via a five-step process, and subsequent *N*-butenylation to give the common intermediate **4t** in 69% yield. Therefore, the present route to **15** using the CuI-catalyzed coupling achieved a 1.6-fold increase in overall yield compared to the prior procedure.

## 3. Conclusions

In this study, a range of 4-alkoxy-1*H*-pyrazoles was synthesized using the CuI-catalyzed coupling reaction of 4-iodopyrazoles with an excess amount of alcohol. Improved syntheses of withasomnine and its homologs were achieved using the products obtained with the present method. The current withasomnine synthetic route was reduced by four steps with a threefold-improvement in the overall yield compared to our previous report [22,23]. However, reducing the amounts of catalysts, ligands, and alcohols will be required to increase the practicality of this reaction in the future.

## 4. Materials and Methods

### 4.1. General Information

NMR spectra were recorded at 27 °C on Agilent 400- and 600-MR-DD2 spectrometers (Agilent Tech., Inc., Santa Clara, CA, USA) in CDCl_3_ with tetramethylsilane (TMS) as an internal standard. HRMS was performed using a JEOL JMS-700 (2) mass spectrometer (JEOL, Tokyo, Japan). Melting points were determined using a Yanagimoto micromelting point apparatus (Yamagimoto, Kyoto, Japan) and were uncorrected. Liquid column chromatography was conducted using silica gel (Fuji Silysia FL-60D). Analytical TLC was performed on precoated Merck glass plates (silica gel 60 F_254_), and compounds were detected by dipping the plates in an ethanol solution of phosphomolybdic acid, followed by heating. All microwave-aided reactions were performed using a Biotage Initiator^®^ (Biotage, Uppsala, Sweden). Allyl alcohol, *n*-propanol, isobutyl alcohol, cyclobutanol, cyclopentanol, cyclohexanol, Phenol, p-methoxyphenol, allylbromide, and Pd(dba)_2_, were purchased from Tokyo Chemical Industry (TCI) Co. (Tokyo, Japan). *t*-BuOK, CuI, 1,10-phenanthroline (**L3**), 3,4,7,8-tetramethyl-1,10-phenanthroline (**L4**), isopropanol, *tert*-butanol, isoamyl alcohol, toluene, cesium acetate, trifluoromethanesulfonic anhydride, and triethylamine were purchased from Nacalai Tesque, Inc. (Kyoto, Japan). Dry xylene, dry DMF, *n*-butanol, *sec*-butanol, and 1,2-dimethoxyethane, were purchased from FUJIFILM Wako Pure Chemical Co. (Osaka, Japan). tBuDavePhos (**L1**), 2-isobutyrylcyclohexanone (**L2**), RuClH(CO)(PPh_3_)_3_, Grubbs^2nd^, and XPhos were purchased from Sigma-Aldrich Co. LLC (St. Louis, MO, USA).

^1^H- and ^13^C-NMR spectra of all new compounds with compound **15** are provided in Appendix A as Appendix A.

### 4.2. CuI-Catalyzed Coupling Reactions of 4-iodo-1H-1-tritylpyrazole with Alcohols (Table 1 and Table 2)

*General procedure* (Table 1, entry 6): To a solution of **2a** (50.0 mg, 0.12 mmol) in allyl alcohol (2.0 mL, 29 mmol) in a microwave vial (0.5–2.0 mL) were added 3,4,7,8-tetramethyl-1,10-phenanthroline (5.8 mg, 0.026 mmol, 20 mol%), CuI (4.4 mg, 0.023 mmol, 20 mol%), and *^t^*BuOK (28.8 mg, 0.26 mmol, 2.0 equiv). The mixture was stirred to make a solution, sealed, and heated at 130 °C for 1 h under MW irradiation. The cooled mixture was checked by TLC (hexane/AcOEt = 8:1), quenched by the addition of saturated (sat.) aqueous (aq.) NH_4_Cl (1 mL), and extracted with dichloromethane (CH_2_Cl_2_; 1.0 mL × 3). The combined organic layers were dried over MgSO_4_, filtered, and evaporated to give a crude residue that was purified by silica gel column chromatography (eluent:hexane/AcOEt = 20:1) to afford previously reported **4a** (27.9 mg, 66%).

**4a**: known [22,23]

4-Methoxy-1*H*-1-tritylpyrazole (**4c**): Colorless needles (CH_2_Cl_2_); mp 149–152 °C; ^1^H-NMR (400 MHz, CDCl_3_): δ 7.40 (1H, s, pyrazole-H), 7.32–7.28 (9H, m, Tr-H), 7.18–7.14 (6H, m, Tr-H), 7.01 (1H, s, pyrazole-H), 3.68 (3H, s, 4-OMe); ^13^C-NMR (100 MHz, CDCl_3_): δ 145.7, 143.2, 130.1, 127.6, 127.4, 127.2, 117.2, 78.6, 58.7; HREIMS *m*/*z* calculated (calcd) for C_23_H_2_0N_2_O (M^+^) 340.1575, found 340.1577.

4-Ethoxy-1*H*-1-tritylpyrazole (**4d**): Colorless needles (CH_2_Cl_2_); mp 141–144 °C; ^1^H-NMR (400 MHz, CDCl_3_): δ 7.40 (1H, s, pyrazole-H), 7.01 (1H, s, pyrazole-H), 7.31–7.28 (9H, m, Tr-H), 7.18–7.14 (6H, m, Tr-H), 3.87 (2H, q *J* = 6.6 Hz, -OC*H*_2_CH_3_), 1.32 (3H, t, *J* = 6.6 Hz, CH_2_C*H*_3_); ^13^C-NMR (100 MHz, CDCl_3_): δ 144.4, 143.2, 130.1, 127.8, 127.6, 127.6, 117.9, 78.6, 67.1, 14.9; HREIMS *m/z* calcd for C_24_H_22_N_2_O (M^+^) 354.1732, found 354.1735.

4-*n*-Propyloxy-1*H*-1-tritylpyrazole (**4e**): Colorless needles (CH_2_Cl_2_); mp 118–120 °C; ^1^H-NMR (400 MHz, CDCl_3_): δ, 7.40 (1H, s, pyrazole-H), 7.30–7.27 (9H, m, Tr-H), 7.18–7.14 (6H, m, Tr-H), 7.01 (1H, s, pyrazole-H), 3.76 (2H, t, *J* = 6.6 Hz, -OC*H*_2_CH_2_-), 1.72 (2H, qt, *J* = 7.4, 6.6 Hz, -CH_2_C*H*_2_CH_3_), 0.97 (3H, t, *J* = 7.4 Hz, -CH_2_C*H*_3_); ^13^C-NMR (100 MHz, CDCl_3_): δ 144.6, 143.2, 130.1, 127.8, 127.61, 127.59, 117.7, 78.5, 73.1, 22.7, 10.4; HREIMS *m*/*z* calcd for C_25_H_24_N_2_O (M^+^) 368.1889, found 368.1889.

4-*n*-Butoxy-1*H*-1-tritylpyrazole (**4f**): Colorless needles (CH_2_Cl_2_); mp 127–130 °C; ^1^H-NMR (400 MHz, CDCl_3_): δ 7.40 (1H, s, pyrazole-H), 7.31–7.28 (9H, m, Tr-H), 7.18–7.14 (6H, m, Tr-H), 7.01 (1H, s, pyrazole-H), 3.80 (2H, q *J* = 6.6 Hz, -OC*H*_2_CH_2_-), 1.68 (2H, quint, *J* = 6.7 Hz, -CH_2_C*H*_2_CH_2_-), 1.42 (2H, sext, *J* = 6.6 Hz, -CH_2_C*H*_2_CH_3_), 0.93 (3H, t, *J* = 6.7 Hz, -CH_2_C*H*_3_); ^13^C-NMR (100 MHz, CDCl_3_): δ 144.7, 143.2, 130.1, 127.8, 127.6, 127.6, 117.7, 78.5, 71.3, 31.4, 19.1, 13.9; HREIMS *m*/*z* calcd for C_26_H_26_N_2_O (M^+^) 382.2035, found 382.2040.

4-Isopropyloxy-1*H*-1-tritylpyrazole (**4g**): White powder; mp 91–93 °C; ^1^H-NMR (400 MHz, CDCl_3_): δ 7.39 (1H, d, *J* = 1.0 Hz, pyrazole-H), 7.31–7.28 (9H, m, Tr-H), 7.18–7.14 (6H, m, Tr-H), 7.02 (1H, d, *J* = 0.7 Hz, pyrazole-H), 4.12 (1H, sept, *J* = 6.1 Hz, -OC*H*(CH_3_)_2_), 1.26 (6H, d, *J* = 6.1 Hz, -CH(C*H*_3_)_2_); ^13^C-NMR (100 MHz, CDCl_3_): δ 143.2, 142.7, 130.1, 129.2, 127.6, 119.8, 78.6, 74.0, 21.9; HREIMS *m*/*z* calcd for C_25_H_24_N_2_O (M^+^) 368.1889, found 368.1885.

4-Isobutoxy-1*H*-1-tritylpyrazole (**4i**): White powder; mp 95–98 °C; ^1^H-NMR (400 MHz, CDCl_3_): δ 7.40 (1H, s, pyrazole-H), 7.30–7.28 (9H, m, Tr-H), 7.18–7.14 (6H, m, Tr-H), 7.02 (1H, s, pyrazole-H), 3.56 (2H, d, *J* = 6.7 Hz, -OC*H*_2_CH-), 1.99 (1H, nonet, *J* = 6.7 Hz, -CH_2_C*H*(CH_3_)_2_), 0.96 (6H, d, *J* = 6.7 Hz, -CH(C*H*_3_)_2_); ^13^C-NMR (100 MHz, CDCl_3_): δ 144.8, 143.2, 130.1, 127.8, 127.65, 127.61, 127.58, 117.6, 78.5, 78.0, 28.4, 19.1; HREIMS *m*/*z* calcd for C_26_H_26_N_2_O (M^+^) 382.2035, found 382.2040.

4-Isoamyloxy-1*H*-1-tritylpyrazole (**4k**): White powder; mp 77–75 °C; ^1^H-NMR (400 MHz, CDCl_3_): δ 7.40 (1H, s, pyrazole-H), 7.31–7.29 (9H, m, Tr-H), 7.18–7.14 (6H, m, Tr-H), 7.01 (1H, s, pyrazole-H), 3.83 (2H, q, *J* = 6.6 Hz, -OC*H*_2_CH_2_-), 1.76 (1H, nonet, *J* = 6.7 Hz, -CH_2_C*H*(CH_3_)_2_), 1.58 (2H, q, *J* = 6.7 Hz, -CH_2_C*H*_2_CH-), 0.92 (6H, d, *J* = 6.7 Hz, -CH (C*H*_3_)_2_); ^13^C-NMR (100 MHz, CDCl_3_): δ 22.6, 24.8, 38.1, 70.0, 78.6, 117.8, 127.6, 127.7, 127.8, 130.1, 143.2, 144.6; HREIMS *m*/*z* calcd for C_26_H_26_N_2_O (M^+^) 396.2201, found 396.2201.

4-Cyclobutyloxy-1*H*-1-tritylpyrazole (**4l**): White powder; mp 119–121 °C; ^1^H-NMR (400 MHz, CDCl_3_): δ 7.34 (1H, br s, pyrazole-H), 7.31–7.28 (9H, m, Tr-H), 7.17–7.12 (6H, m, Tr-H), 6.95 (1H, br s, pyrazole-H), 4.40–4.33 (1H, m, -OC*H*(CH_2_)_2_-), 2.31–2.33 (2H, m), 2.13–2.03 (2H, m), 1.81–1.73 (1H, m), 1.62–1.52 (1H, m); ^13^C-NMR (100 MHz, CDCl_3_): δ 143.2, 142.4, 130.1, 128.3, 127.6, 118.6, 78.6, 74.4, 30.3, 12.6; HREIMS *m*/*z* calcd for C_26_H_24_N_2_O (M^+^) 380.1889, found 380.1890.

4-Cyclopentyloxy-1*H*-1-tritylpyrazole (**4m**): White powder; mp 104–106 °C; ^1^H-NMR (400 MHz, CDCl_3_): δ 7.37 (1H, d, *J* = 0.6 Hz, pyrazole-H), 7.37–7.28 (9H, m, Tr-H), 7.17–7.12 (6H, m, Tr-H), 6.98 (1H, d, *J* = 0.8 Hz, pyrazole-H), 4.44–4.42 (1H, m, -OC*H*(CH_2_)_2_-), 1.82–1.74 (4H, m), 1.72–1.56 (2H, m); ^13^C-NMR (100 MHz, CDCl_3_): δ 143.2, 130.1, 128.7, 127.8, 127.7, 127.6, 119.0, 83.1, 78.5, 32.6, 23.8; HREIMS *m*/*z* calcd for C_27_H_26_N_2_O (M^+^) 394.2045, found 394.2043.

4-Cyclohexyloxy-1*H*-1-tritylpyrazole (**4n**): White powder; mp 132–135 °C; ^1^H-NMR (400 MHz, CDCl_3_): δ 7.39 (1H, s, pyrazole-H), 7.31–7.28 (9H, m, Tr-H), 7.17–7.12 (6H, m, Tr-H), 7.03 (1H, s, pyrazole-H), 3.78–3.83 (1H, m, -OC*H*(CH_2_)_2_-), 1.95–1.92 (2H, m), 1.57–1.51 (2H, m), 1.48–1.38 (2H, m) 1.32–1.23 (2H, m); ^13^C-NMR (100 MHz, CDCl_3_): δ 143.2, 142.5, 130.1, 129.5, 127.6, 120.1, 79.6, 78.6, 31.8, 25.6, 23.6; HREIMS *m*/*z* calcd for C_28_H_28_N_2_O (M^+^) 408.2202, found 408.2201.

4-Benzyloxy-1*H*-1-tritylpyrazole (**4o**): White powder; mp 154–156 °C; ^1^H-NMR (400 MHz, CDCl_3_): δ 7.44 (1H, s, pyrazole-H), 7.35–7.28 (14H, m, Tr-H, Ph-H), 7.15–7.12 (6H, m, Tr-H, Ph-H), 7.02 (1H, s, pyrazole-H), 4.86 (2H, br s, -OC*H*_2_Ph); ^13^C-NMR (100 MHz, CDCl_3_): δ 144.1, 143.1, 136.7, 130.1, 128.5, 128.0, 127.8, 127.6, 118.7, 78.6, 73.7; HREIMS *m*/*z* calcd for C_29_H_24_N_2_O (M^+^) 416.1889, found 416.1889.

(*E*/*Z*)-4-(Allyloxy)-1-(prop-1-en-1-yl)-1*H*-pyrazole (**4r**): Colorless oil; ^1^H-NMR (400 MHz, CDCl_3_): δ 7.38 (0.1H, s, pyrazole-H), 7.32 (0.7H, br s, pyrazole-H), 7.28 (0.3H, d, *J* = 0.6 Hz, pyrazole-H), 7.23 (0.7H, d, *J* = 0.8 Hz, pyrazole-H), 6.74 (0.7H, dq, *J* = 14.2, 0.7 Hz, (*E*)-NC*H*=CHCH_3_), 6.68 (0.3H, dq, *J* = 9.4, 0.7 Hz, (*Z*)-NC*H*=CHCH_3_), 6.08–5.97 (1H, m, -OCH_2_C*H*=CH_2_), 5.85 (0.7H, dq, *J* = 14.2, 7.0 Hz, (*E*)-NCH=C*H*CH_3_), 5.40 (1H, br d, *J* = 17.2 Hz, -CH=C*H*H), 5.29 (1H, br d, *J* = 9.4 Hz, -CH=CH*H*), 5.26 (0.3H, dq, *J* = 9.4, 7.9, (*Z*)-NCH=C*H*CH_3_), 4.44 (0.6H, dt, *J* = 5.5, 1.2Hz, -OC*H*_2_CH=CH_2_), 4.42 (1.4H, dt, *J*
*=* 5.5, 1.5 Hz, -OC*H*_2_CH=CH_2_), 1.95 (0.9H, dd, *J* = 7.4, 1.8 Hz, (*Z*)-NCH=CHC*H*_3_), 1.81 (2.1H, dd, *J* = 7.1, 0.6 Hz, (*E*) -NCH=CHC*H*_3_); ^13^C-NMR (100 MHz, CDCl_3_): δ 146.0, 133.1, 133.0, 128.8, 128.72, 127.67, 127.65, 118.1, 115.0, 111.8, 111.1, 72.55, 72.47,, 14.7, 12.8 (3 carbons are overlapped); HREIMS *m*/*z* calcd for C_9_H_12_N_2_O (M^+^) 164.0950, found 164.0949.

**4s**: known [24]

4-Allyloxy-1-(3-buten-1-yl)pyrazole (**4t**): Colorless oil; ^1^H-NMR (400 MHz, CDCl_3_): δ 7.24 (1H, d, *J* =1.2 Hz, pyrazole-H), 7.08 (1H, d, *J* =1.0 Hz, pyrazole-H), 6.01 (1H, ddt, *J* = 17.2, 10.5, 5.5 Hz, -OCH_2_C*H*=CH_2_), 5.74 (1H, ddt, *J* = 17.2, 10.4, 6.8 Hz, -CH_2_C*H*=CH_2_), 5.38 (1H, dq, *J* = 17.2, 1.6 Hz, -CH_2_CH=C*H*H), 5.28 (1H, dq, *J* = 10.4, 1.4 Hz,-CH_2_CH=CH*H*), 5.04–5.10 (2H, overlapped, 2 × -CH=C*H*H), 4.40 (2H, dt, *J* = 5.4, 1.5 Hz, -OC*H_2_*CH=), 4.07 (2H, t, *J* = 7.1 Hz, NC*H*_2_CH_2_-), 2.57 (2H, qt, *J* = 7.0, 1.2 Hz, -CH_2_C*H*_2_CH=CH_2_); ^13^C-NMR (100 MHz, CDCl_3_): δ144.9, 134.1, 133.3, 127.0, 117.8, 117.4, 115.0, 72.5, 52.1, 34.5; HREIMS *m*/*z* calcd for C_10_H_14_N_2_O (M^+^) 178.1106, Found 178.1105.

### 4.3. Modified Synthesis of Withasomnine, (Scheme 2)

#### 4.3.1. Synthesis of 1-allyl-4-iodo-1*H*-pyrazole (**2c**) 

To a solution of 4-iodopyrazole **1** (500.0 mg, 2.6 mmol) in acetone (5 mL), 20% NaOH aq. (0.5 mL, 1.5 equiv) was added with stirring followed by allyl bromide (0.2 mL, 3.9 mmol, 1.5 equiv). The reaction mixture was stirred at room temperature for 1 h. After checking by TLC (hexane/AcOEt = 2:1), sat. aq. NH_4_Cl (5 mL) was added to the reaction mixture to quench the reaction. The mixture was extracted with CH_2_Cl_2_ (10 mL × 3) and the combined organic layers were washed with brine (5 mL × 2), dried over MgSO_4_, and filtered. The solvent was removed under reduced pressure to give a crude residue that was purified by silica gel column chromatography (eluent:hexane/AcOEt = 2:1) to afford **2c** (566.2 mg, 96%): colorless oil; ^1^H-NMR (400 MHz, CDCl_3_): δ 7.53 (1H, s, pyrazole-H), 7.45 (1H, s, pyrazole-H), 6.00 (1H, ddt, *J* = 17.1, 10.2, 6.1 Hz, -NHCH_2_C*H*=CH_2_), 5.30 (1H, dq, *J* = 10.2, 1.2 Hz, -CH_2_CH=C*H*H), 5.26 (1H, dq, *J* = 17.1, 1.4 Hz, -CH_2_CH=CH*H*), 4.75 (2H, dt, *J* = 6.3, 1.4 Hz, -NC*H*_2_CH=CH_2_); ^13^C-NMR (100 MHz, CDCl_3_) δ 144.4, 133.4, 132.2, 119.3, 56.2, 55.1; HREIMS *m*/*z* calcd for C_6_H_7_N_2_I (M^+^) 233.9654, found 233.9653.

#### 4.3.2. Synthesis of (*E*/*Z*)-4-iodo-1-(prop-1-en-1-yl)-1*H*-pyrazole (**2b**)

To a MW vial containing a solution of **2c** (571.5 mg, 2.4 mmol) in toluene (2 mL) was added the ruthenium hydride catalyst, RuClH(CO)(PPh_3_)_3_ (116.3 mg, 0.12 mmol, 5 mol%). The reaction mixture in the sealed vial was heated at 160 °C for 10 min under MW irradiation. After removal of the solvent from the mixture, the residue was purified by silica gel column chromatography (eluent:hexane/AcOEt = 10:1) to afford **2b** (546.6 mg, 96%) as a colorless oil: ^1^H-NMR (400 MHz, CDCl_3_): δ 7.62 (0.2H, s, pyrazole-H), 7.61 (0.2H, s, pyrazole-H), 7.58 (0.8H, s, pyrazole-H), 7.55 (0.8H, s, pyrazole-H), 6.80 (0.8H, dq, *J* = 14.0, 1.6 Hz, -C*H_E_*=CHCH_3_), 6.76 (0.2H, dq, *J* = 9.2, 1.8 Hz, -C*H_Z_*=CHCH_3_), 6.04 (0.8H, dq, *J* = 13.8, 7.0 Hz, -CH=C*H_E_*CH_3_), 5,45 (0.2H, br quint, *J* = 7.3 Hz, -CH=C*H*_*Z*_CH_3_), 1.94 (0.4H, dd, *J* = 7.4, 1.8 Hz, -CH=CHC*H*_3_), 1.82 (2.6H, dd, *J* = 6.9, 1.8 Hz, -CH=CHC*H*_3_); ^13^C-NMR (100 MHz, CDCl_3_): δ 145.0, 144.8, 134.0, 133.8, 133.6, 131.3, 128.7, 128.5, 128.4, 127.4, 126.7, 116.7, 114.1, 57.6, 57.4, 14.7, 12.9; HREIMS *m*/*z* calcd for C_6_H_7_N_2_I (M^+^) 233.9654, found 233.9652.

#### 4.3.3. Synthesis of (*E*/*Z*)-5-allyl-1-(prop-1-en-1-yl)-1*H*-pyrazol-4-yl trifluoromethane-sulfonate (**12**)

A solution of **4r** (418.9 mg, 2.6 mmol) in 1,2-dimethoxethane (DME, 2 mL) in a sealed vial was heated at 180 °C for 30 min under MW irradiation. The reaction mixture was concentrated directly under reduced pressure to give a crude residue that was purified by silica gel column chromatography (eluent:hexane/AcOEt = 2:1) to give (*E*/*Z*)-**12** (364.3 mg, 87%) as a colorless oil. ^1^H-NMR (400 MHz, CDCl_3_): δ 7.30 (0.3H, s, pyrazole-H), 7.24 (0.7H, s, pyrazole-H), 6.62 (0.7H, dd, *J* = 13.7, 1.6 Hz, (*E*)-NC*H*=CHCH_3_), 6.50 (0.3H, dd, *J* = 8.7, 1.6 Hz, (*Z*)-NC*H*=CHCH_3_), 6.15 (0.7H, dq, *J =* 13.9, 6.9 Hz, (*E*)-CH=C*H*CH_3_), 5.83–5.94 (1H, m, -CH_2_C*H*=CH_2_), 5.54 (0.3H, dq, *J* = 8.7, 7.4 Hz, (*Z*)-CH=C*H*CH_3_), 5.12–5.17 (1H, m, -CH_2_CH=C*H*H), 5.00 (1H, br, -OH), 5.04 (1H, ddd, *J* = 17.0, 7.2, 1.1 Hz, -CH_2_CH=CH*H*), 3.43 (1.4H, d, *J* = 5.7 Hz, -C*H*_2_CH=CH_2_), 3.38 (0.6H, d, *J* = 6.1 Hz, -C*H*_2_CH=CH_2_), 1.87 (0.9H, dd, *J* = 7.2, 1.4 Hz, -CH=CHC*H*_3_), 1.80 (2.1H, dd, *J* = 6.9, 1.2 Hz, -CH=CHC*H*_3_); ^13^C-NMR (100 MHz, CDCl_3_): δ 139.2, 138.8, 133.5, 133.3, 129. 4, 129.1, 126.9, 125.8, 124.8, 124.6, 123.4, 116.7, 116.5, 114.8, 27.0, 26.5, 15.1, 12.9; HREIMS *m*/*z* calcd for C_9_H_12_N_2_O (M^+^) 164.0950, found 164.0949.

#### 4.3.4. Synthesis of (*E*/*Z*)-5-allyl-1-(prop-1-en-1-yl)-1*H*-pyrazol-4-yl trifluoromethanesulfonate (**13**)

To a solution of **12** (264.7 mg, 1.6 mmol) in CH_2_Cl_2_ (4 mL) was added triethylamine (0.3 mL, 2.4 mmol, 1.5 equiv) at −20 °C with stirring. After stirring for 10 min, Tf_2_O (0.4 mL, 2.4 mmol, 1.5 equiv) was added dropwise to the reaction mixture. After stirring at room temperature for another 1 h, the reaction was quenched by the addition of sat. aq. NH_4_Cl (5 mL) and extracted with CH_2_Cl_2_ (5 mL × 3). The combined organic layers were washed with brine (5 mL × 2), dried over MgSO_4_, filtered, and evaporated. The obtained residue was purified by silica gel column chromatography (eluent:hexane/AcOEt = 2:1) to give **13** (430.4 mg, 90%) as an (*E/Z*) mixture. (*Z*)-**13**: Colorless oil; ^1^H-NMR (600 MHz, CDCl_3_): δ 7.58 (1H, s, pyrazole-H), 6.54 (1H, dq, *J* = 8.8, 1.8 Hz, -C*H*=CHCH_3_), 5.79 (1H, ddt, *J* = 17.0, 10.0, 5.9 Hz, -CH_2_C*H*=CH_2_), 5.71 (1H, dq, *J* = 8.8, 7.3 Hz, -CH=C*H*CH_3_), 5.17 (1H, br dq, *J* = 10.3, 1.8 Hz, -CH_2_CH=C*H*H), 5.06 (1H, br dq, *J* = 17.0, 1.8 Hz, -CH_2_CH=CH*H*), 3.42 (2H, dt, *J* = 6.1, 1.6 Hz, -C*H*_2_CH=CH_2_), 1.87 (3H, dd, *J* = 7.4, 1.8 Hz, -CH=CHC*H*_3_); ^13^C-NMR (100 MHz, CDCl_3_): δ 132.4, 131.4, 131.2, 130.9, 124.4, 123.9, 118.7 (q, *J* = 321.43 Hz, -CF_3_), 118.2, 27.2, 12.9; HREIMS *m*/*z* calcd for C_10_H_11_F_3_N_2_O_3_S (M^+^) 296.0442, found 296.0443. (*E*)-**13**: colorless oil; ^1^H-NMR (600 MHz, CDCl_3_): δ 7.58 (1H, s, pyrazole-H), 6.54 (1H, dq, *J* = 8.8, 1.8 Hz, -C*H*=CHCH_3_), 5.79 (1H, ddt, *J* = 17.0, 10.0, 5.9 Hz, -CH_2_C*H*=CH_2_), 5.83 (1H, dq, *J* = 8.8, 7.3 Hz, -CH=C*H*CH_3_), 5.21 (1H, br dq, *J* = 10.3, 1.7 Hz, -CH_2_CH=C*H*H) 5.06 (1H, br dq, *J* = 17.0, 1.8 Hz, -CH_2_CH=CH*H*), 3.47 (2H, dt, *J* = 5.9, 1.8 Hz, -C*H*_2_CH=CH_2_), 1.84 (3H, dd, *J* = 7.0, 1.9 Hz, -CH=CHC*H*_3_); ^13^C-NMR (100 MHz, CDCl_3_): δ 131.6 131.4, 131.0, 130.9, 124.3, 118.7 (q, *J* = 321.3Hz, -CF_3_), 118.3, 117.9, 26.7, 15.0; HREIMS *m*/*z* calcd for C_10_H_11_F_3_N_2_O_3_S (M^+^) 296.0442, found 296.0443.

#### 4.3.5. Synthesis of 4*H*-pyrrolo [1,2-*b*]pyrazol-3-yl trifluoromethanesulfonate (**14**)

To a solution of **13** (50.0 mg, 0.17 mmol) in CH_2_Cl_2_ (2 mL) were added Grubbs^2nd^ catalyst (7.1 mg, 0.0085 mmol, 5 mol%) and CuI (0.8 mg, 0.0042 mmol, 2.5 mol%). The sealed reaction mixture was heated at 80 °C for 1 h under MW irradiation. The solvent was removed from the mixture under reduced pressure to give a crude material that was purified by silica gel column chromatography (eluent:hexane/AcOEt = 4:1) to afford **14** (27.0 mg, 63%). **14**: Colorless oil; ^1^H-NMR (400 MHz, CDCl_3_): δ 7.55 (1H, s, pyrazole-H), 7.21 (1H, dt, 4.1, 2.2 Hz, -CH_2_CH=C*H*-), 6.08 (1H, m, -CH_2_C*H*=CH-), 3.55 (2H, br t, *J* = 2.3 Hz, ArC*H*_2_CH=CH-); ^13^C-NMR (100 MHz, CDCl_3_): δ 135.0, 134.0, 130.0, 129.0, 118.6 (q, *J* = 322.0 Hz, -CF_3_), 118.4, 30.3; HREIMS *m*/*z* calcd for C_7_H_5_F_3_N_2_O_3_S (M^+^) 253.9973, found 253.9977.

#### 4.3.6. Synthesis of 5,6-dihydro-4*H*-pyrrolo[1,2-*b*]pyrazol-3-yl trifluoromethanesulfonate (**6**)

To a solution **14** (50.0 mg, 0.20 mmol) in MeOH (5 mL) was added Pd/C (5.00 mg, 10 mol%). The mixture was stirred for 24 h at room temperature under hydrogen gas at 1 atm. After removal of Pd/C by filtration, the solvent was evaporated to give a crude mixture that was purified by silica gel column chromatography (eluent:hexane/AcOEt = 8:1) to afford **6** (45.6 mg, 90%) [22,23].

### 4.4. Synthesis of Withasomnine Homolog **15** (Scheme 3)

#### 4.4.1. Synthesis of 1-allyl-4-iodo-1*H*-pyrazole (**2d**)

To a solution of **1** (300.0 mg, 1.5 mmol) in acetone (9 mL) was added 20% NaOH aq. (6 mL, excess) with stirring, followed by allyl bromide (0.3 mL, 3.0 mmol), and the reaction mixture was stirred at rt for 1 h. After quenching with sat. aq. NH_4_Cl (10 mL), the mixture was extracted with CH_2_Cl_2_ (20 mL × 3), and the combined organic layers were washed with brine (5 mL × 2), dried over MgSO_4_, and filtered. The solvent was removed under reduced pressure to give a crude residue that was purified by silica gel column chromatography (eluent:hexane/AcOEt = 3:1) to afford **2d** (373.9 mg, 88%). **2d**: Colorless oil;^1^ H-NMR (400 MHz, CDCl_3_): δ 7.50 (1H, s, pyrazole-H), 7.42 (1H, s, pyrazole-H), 5.72 (1H, ddt, *J* = 17.4, 9.9, 7.0 Hz, -CH_2_C*H*=CH_2_), 5.08 (1H, br d, *J* = 17. 4 Hz, -CH=C*H*H), 5.08 (1H, br d, *J* = 9.9 Hz, -CH=CH*H*), 4.18 (2H, t, *J* = 7.0 Hz, NHC*H*_2_CH_2_-), 2.60 (2H, br q, *J* = 7.0 Hz, -CH_2_C*H*_2_CH=); ^13^C-NMR (100 MHz, CDCl_3_) δ 144.2, 133.7, 133.5, 117.9, 55.6, 52.0, 34.5 (one aromatic or olefinic carbon signal is overlapped); HREIMS *m*/*z* calcd for C_7_H_9_N_2_I (M^+^) 247.9810, found 247.9810.

#### 4.4.2. Synthesis of 5-allyl-1-(but-3-en-1-yl)-4-hydroxy-1*H*-pyrazole (**16**)

A solution of **4t** (81.8 mg, 0.46 mmol) in DME (2 mL) in a sealed vial was heated at 200 ^o^C for 1 h under MW irradiation. The reaction mixture was concentrated directly under reduced pressure to give a crude residue that was purified by silica gel column chromatography (eluent:hexane/AcOEt = 2:1) to give **16** (70.0 mg, 81%). **16**: Colorless oil; ^1^H-NMR (400 MHz, CDCl_3_): δ 7.12 (1H, s, pyrazole-H), 6.91 (1H, br s, -OH), 5.87 (1H, ddt, *J* = 17.0, 10.0, 5. 8 Hz, -OCH_2_C*H*=CH_2_), 5.71 (1H, ddt, *J* = 17.0, 10.4, 7.0, -CH_2_C*H*=CH_2_), 5.11 (1H, dq, *J* = 10.2, 1.4 Hz, -CH=C*H*H), 5.01–5.07 (3H, overlapped. 3 × -CH=C*H*H), 3.98 (2H, t, *J* = 7.4 Hz, NC*H*_2_CH_2_-), 3.39 (2H, dt, *J* = 5.8, 1.5 Hz, ArC*H_2_*CH=), 2.49 (2H, br q, *J* = 7. 2 Hz, -CH_2_C*H*_2_CH=); ^13^C-NMR (100 MHz, CDCl_3_): δ 138.3, 134.1, 133.7, 127.6, 126.0, 116.6, 116.4, 48.9, 34.5, 27.0; HREIMS *m*/*z* calcd for C_10_H_14_N_2_O (M^+^) 178.1106, Found 178.1104.

#### 4.4.3. Synthesis of 5-allyl-1-(but-3-en-1-yl)-1*H*-pyrazol-4-yl trifluoromethanesulfonate (**17**)

To a solution of **16** (70.0 mg, 0.39 mmol) in CH_2_Cl_2_ (10 mL) was added triethylamine (0.06 mL, 0.43 mmol, 1.1 equiv) at −20 °C with stirring. After stirring for 10 min, Tf_2_O (0.1 mL, 0.6 mmol, 1.5 equiv) was added dropwise to the reaction mixture. After stirring at room temperature for another 1 h, the reaction was quenched by the addition of sat. aq. NH_4_Cl (5 mL) and extracted with CH_2_Cl_2_ (10 mL × 3). The combined organic layers were washed with brine (5 mL × 2), dried over MgSO_4_, filtered, and concentrated. The obtained residue was purified by silica gel column chromatography (eluent:hexane/AcOEt = 2:1) to give **17** (100.6 mg, 83%). **17**: Colorless oil; ^1^H-NMR (400 MHz, CDCl_3_): δ 7.48 (1H, s, pyrazole-H), 5.88–5.67 (2H, overlapped, 2 × -CH_2_C*H*=CH_2_), 5.20 (1H, dq, *J* = 10.2, 1.2 Hz, -CH=C*H*H), 5.12–5.03 (3H, overlapped. 3 × -CH=C*H*H), 4.06 (2H, t, *J* = 7.3 Hz, NC*H*_2_CH_2_-), 3.44 (2H, dt, *J* = 5.8, 1.6 Hz, ArC*H_2_*CH=), 2.57 (2H, br q, *J* = 7. 4 Hz, -CH_2_C*H*_2_CH=); ^13^C-NMR (100 MHz, CDCl_3_): δ 133.2, 131.5, 130.6, 129.7, 118.4 (q, *J*_C-F_ = 321.2 Hz), 117.8, 117.7, 100.5, 49.4, 33.8, 26.9; HREIMS *m*/*z* calcd for C_11_H_13_F_3_N_2_O_3_S (M^+^) 310.0599, Found 310.0598.

#### 4.4.4. Synthesis of 7,8-dihydro-4*H*-pyrazolo[1,5-*a*]azepin-3-yl trifluoromethanesulfonate (**18**)

To a solution of **17** (99.1 mg, 0.32 mmol) in CH_2_Cl_2_ (2 mL) were added Grubbs^2nd^ catalyst (13 mg, 0.016 mmol, 5 mol%) and CuI (1.0 mg, 0.005 mmol, 2.5 mol%). The sealed reaction mixture was heated at 80 °C for 1 h under MW irradiation. Solvent was removed under reduced pressure to give a crude mixture that was purified by silica gel column chromatography (eluent:hexane/AcOEt = 4:1) to afford **18** (65.1 mg, 72%). **18**: Amorphous solid; mp 72–75 °C; ^1^H-NMR (400 MHz, CDCl_3_): δ 7.38 (1H, s, pyrazole-H), 5.77–5.70 (2H, m, 2 × -C*H*=CHCH_2_-), 4.46 (2H, dd, *J* = 5.6, 4.5 Hz, NC*H*_2_CH_2_-), 3.47–3.45 (2H, m, ArC*H*_2_CH=), 2.49–2.45 (2H, m, -CH_2_C*H_2_*CH=); ^13^C-NMR (100 MHz, CDCl_3_): δ 134.0, 129.8, 129.16, 129.12, 122.4, 118.6 (q, *J*_C-F_ = 321.2 Hz), 50.9, 27.8, 21.7; HREIMS *m*/*z* calcd for C_9_H_9_F_3_N_2_O_3_S (M^+^) 282.0286, found 282.0282.

#### 4.4.5. Synthesis of 3-phenyl-7,8-dihydro-4*H*-pyrazolo[1,5-*a*]azepine (**19**)

To a solution of **18** (54.7 mg, 0.19 mmol) in DME/H_2_O = 9:1 (5 mL) in a MW vial were added XPhos (9.2 mg, 0.019 mmol, 10 mol%), Pd(dba)_2_ (11.0 mg, 0.019 mmol, 10 mol%), cesium carbonate (126.4 mg, 0.38 mmol, 2.0 eq.), and phenylboronic acid (50.0 mg, 0.41 mmol, 2.0 eq.). The sealed vial was heated under MW irradiation at 130 °C for 1 h. The cooled reaction mixture was quenched with sat. aq. NH_4_Cl solution (40 mL) and extracted with CH_2_Cl_2_ (10 mL × 3). The combined organic layers were dried over MgSO_4_, filtered, and the solvent was removed under reduced pressure to give a crude residue that was purified by silica gel column chromatography (eluent:hexane/AcOEt = 2:1) to afford **19** (35.4 mg, 87%). **19**: Oil; ^1^H-NMR (400 MHz, CDCl_3_): δ 7.49 (1H, s, pyrazole-H), 7.42–7.34 (2H, m, Ph-H), 7.34–7.26 (3H, m, Ph-H), 5.73–5.56 (2H, m, -NCH_2_CH=CH-, -CH=CHCH_2_-), 4.54–4.51 (m, 2H), 3.61–3.59 (2H, m), 2.50–2.46 (2H, m); ^13^C-NMR (100 MHz, CDCl_3_): δ 138.9, 136.8, 133.6, 129.1, 128.9, 128.6, 128.2, 126.3, 123.8, 120.7, 49.5, 28.4, 23.1; HREIMS *m*/*z* calcd for C_14_H_14_N_2_ (M^+^) 210.1157, found 210.1156.

#### 4.4.6. Synthesis of 3-phenyl-5,6,7,8-tetrahydro-4*H*-pyrazolo[1,5-*a*]azepine (**15**)

To a solution of **19** (3.0 mg, 0.014 mmol) in MeOH (10 mL) was added Pd/C (0.2 mg, 10 mol%). The mixture was stirred overnight at room temperature under hydrogen gas at 1 atm. After removal of Pd/C by filtration, the solvent was evaporated to give a crude residue that was purified by column chromatography (eluent:hexane/AcOEt = 4:1) to afford **15** (2.8 mg, 92%). **15**: ^1^H-NMR (400 MHz, CDCl_3_): δ 7.44 (1H, s, pyrazole-H), 7.37–7.42 (2H, m, Ph-H), 7.33–7.25 (3H, m, Ph-H), 4.35–4.30 (2H, m, -NC*H*_2_CH_2_-), 2.91–2.86 (2H, m, ArC*H*_2_CH_2_-), 1.93–1.86 (2H, m), 1.86–1.80 (2H, m), 1.74–1.67 (2H, m); lit. ^1^H-NMR (400 MHz, CDCl_3_): δ 7.43 (1H, s, pyrazole-H), 7.40–7.26 (5H, m, Ph-H), 4.33–4.31 (2H, m, -NC*H*_2_CH_2_-), 2.89–2.86 (2H, m, ArC*H*_2_CH_2_-) 1.89–1.79 (4H, m), 1.75–1.67 (2H, m) [26]; ^13^C-NMR (100 MHz, CDCl_3_): δ 140.7, 136.6, 134.1, 128.6, 128.3, 126.1, 121.0, 53.4, 31.0, 28.0, 26.8, 24.4; lit. ^13^C-NMR (100 MHz, CDCl_3_): δ 140.7, 136.5, 134.1, 128.5, 128.3, 126.1, 121.0, 53.3, 30.9, 28.0, 26.8, 24.4; HREIMS *m*/*z* calcd for C_14_H_16_N_2_ (M^+^) 212.1313, found 212.1312 [26].

#### 4.4.7. Synthesis of **4t** from **4b**

To a solution of **4b** (48.6 mg, 0.39 mmol) in acetone (1 mL) was added 20% NaOH aq. (0.12 mg, 1.5 eq.) with stirring, followed by 1-bromo-3-butene (0.08 mL, 0.08 mmol, 2 eq.). The reaction mixture was heated at 80 °C for 10 min under MW irradiation. After addition of sat. aq. NH_4_Cl (1 mL) to the reaction mixture, it was extracted with CH_2_Cl_2_ (10 mL × 3) and the combined organic layers were washed with brine (5 mL × 2), dried over MgSO_4_, and filtered. The solvent was removed under reduced pressure to give a crude residue that was purified by silica gel column chromatography (eluent:hexane/AcOEt = 3:1) to afford **4t** (48.1 mg, 69%).

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
