# Peer review of "CuI-Catalyzed Coupling Reactions of 4-Iodopyrazoles and Alcohols: Application toward Withasomnine and Homologs"

_molecules, 2021, doi:10.3390/molecules26113370_

Round 1
Reviewer 1 Report
The submitted manuscript reports Cu-catalyzed coupling reactions of 4-iodopyrazoles and alcohols. The manuscript is well written, the experiments are straightforward, and the results are well presented. The work is recommended for publication in Molecules after the minor corrections and modifications listed below.
The Cu-catalyzed coupling reaction of 4-iodopyrazoles and alcohols/phenols was previously reported (Adv. Synth. Catal., 2010, 352, 3431; PLoS ONE, 2012, 7(10): e46712). These references should be included in revised manuscript.
Page 4, Table 2, entry 20: R2 should be H, since water was used.
Page 7, General procedure: The amount of reagents and catalysts (mmol) should be expressed in decimal form (0.12 mmol, 0.26 mmol etc.).
Scheme 2/Scheme 3 and Materials and Methods: The yields for the compounds 2c and 2d are different from those presented in the Schemes 2 and 3.
Author Response
- The Cu-catalyzed coupling reaction of 4-iodopyrazoles and alcohols/phenols was previously reported (Adv. Synth. Catal., 2010, 352, 3431; PLoS ONE, 2012, 7(10): e46712). These references should be included in revised manuscript.
Response: Thank you for information. Two references were added as ref 13,14.
- Page 4, Table 2, entry 20: R2 should be H, since water was used;
Response: OH was revised to H.
- Page 7, General procedure: The amount of reagents and catalysts (mmol) should be expressed in decimal form (0.12 mmol, 0.26 mmol etc.).
Response: we revised as reviewer’s suggestion.
- Scheme 2/Scheme 3 and Materials and Methods: The yields for the compounds 2c and 2d are different from those presented in the Schemes 2 and 3.
Response; We revised Schemes 2 and 3 as pointed.
Reviewer 2 Report
In this paper, the authors achieved direct 4-alkoxylation of 4-iodo-1H-pyrazoles with alcohols by a CuI-catalyzed coupling protocol and obtained the optimal reaction conditions by a series of controlled experiments. Furthermore, the studied method can be applied directly to the synthesis of withasomnine and its six- and seven-membered cyclic homologs, which play a remarkable role in medicine mechanism research, organic chemistry, biomedicine and so on. The topic of this review is very interesting and the viewpoints are legible. In addition, this work has good practical value in application. Based on the reviewer’s opinion, this work can be accepted after major reversion.
- The author should re-organize the Scheme 3.
- some formats and control experiments were still inadequate and inconsistency, which could be further revised and improved.
- From the comparison of entry 6 and 7 of Table 1, we know that extending the time within 0.5-1 h can greatly improve the yield. Why not extend the time on the basis of 1 h, such as 0.5 h, to observe the change of yield?
- The last line “1h” of Table 1 is not standard than the first several lines.
- In conclusion, the author mentioned that “reducing the amounts of catalysts, ligands, and alcohols will be required to increase the practicality of this reaction in the future”. Whether reducing the amount of catalyst, ligand and alcohol will have a serious impact on the yield ?
Author Response
1. The author should re-organize the Scheme 3.
Response: Thank you for suggestion. we modified Scheme 3.
2. some formats and control experiments were still inadequate and inconsistency, which could be further revised and improved.
Response: we modified General procedure of the CuI catalyzed coupling reaction and some other experiments.
3. From the comparison of entry 6 and 7 of Table 1, we know that extending the time within 0.5-1 h can greatly improve the yield. Why not extend the time on the basis of 1 h, such as 0.5 h, to observe the change of yield? 
Response: Thank you for question. We use microwave irradiator in order to reduce reaction time. Then, maximal MW reaction time is set to 1 h in our laboratory. This is from only our laboratory’s convenience. Please take a look on entries 4 and 5. Both of these reactions in sealed tubes between conventional heating reaction (overnight) and MW aided reaction (1 h) at similar conditions resulted relative chemical yields. Therefore, we deduced extended reaction time might not affect dramatic change in chemical yields. Instead of extended reaction time, we examined reaction at higher temperature, but it afforded lower yield of desired product (table 1, entry 8).
4. The last line “1h” of Table 1 is not standard than the first several lines.
Response: we revised the last line as pointed.
5. In conclusion, the author mentioned that “reducing the amounts of catalysts, ligands, and alcohols will be required to increase the practicality of this reaction in the future”. Whether reducing the amount of catalyst, ligand and alcohol will have a serious impact on the yield ?
Response: Yes, this is optimal condition at this time. Reducing amount of catalyst, ligand must lead lowering chemical yield. And this is weak point of this reaction for further exploration. We wrote about that.
Reviewer 3 Report
Manuscript Number: molecules-1235400
entitled: CuI-Catalyzed Coupling Reactions of 4-Iodopyrazoles and Alcohols: Application toward Withasomnine and Homologs
I had great pleasure reviewing this article. This is a well-conducted scientific study, done thoroughly and expressed concisely. Therefore, the manuscript is suitable for Molecules in the present form.
Author Response
I had great pleasure reviewing this article. This is a well-conducted scientific study, done thoroughly and expressed concisely. Therefore, the manuscript is suitable for Molecules in the present form.
Response: Thank you so much for your attention to this work.
Round 2
Reviewer 2 Report
I suggest to accept and publish the present manuscript.